

# Simulation of Heterogeneous Photooxidation of SO₂ and NOₓ in the presence of Gobi Desert Dust Particles under Ambient Sunlight

Zechen Yu and Myoseon Jang

P.O.Box116450, Department of Environmental Engineering Sciences, Engineering School of Sustainable
Infrastructure and Environment, University of Florida, Gainesville, FL, USA, 32611

*Correspondence to*: Myoseon Jang (mjang@ufl.edu)

## Abstract

To improve the simulation of the heterogeneous oxidation of $SO_2$ and $NO_x$ in the presence of authentic mineral dust particles under ambient environmental conditions, the explicit kinetic mechanism was constructed in Atmospheric Mineral Aerosol Reaction (AMAR) model. The formation of sulfate and nitrate was divided into three phases: gas phase, non-dust aqueous phase and dust phase. Specially, AMAR established the mechanistic role of dust chemical characteristics (e.g., photoactivation, hygroscopicity, and buffering capacity) on heterogeneous chemistry. The photo-activation kinetic process of different dust particles was built into the model by measuring the photodegradation rate constant of an impregnated surrogate (malachite green dye) on a dust filter sample (e.g., Arizona Test dust (ATD) and Gobi Desert dust (GDD)) using an online reflective UV-visible spectrometer. The photoactivation parameters were integrated with the heterogeneous chemistry to predict OH radical formation on dust surfaces. A mathematical equation for the hygroscopicity of dust particles was also included in the AMAR model to process the multiphase partitioning of tracers and in-particle chemistry. The buffering capacity of dust, which is related to the neutralization of dust alkaline carbonates with inorganic acids, was included in the model to dynamically predict the hygroscopicity of aged dust. The AMAR model simulated the formation of sulfate and nitrate using experimental data obtained in the presence of authentic mineral dust under ambient sunlight using a large outdoor smog chamber (UF-APHOR). Overall, both GDD and ATD significantly enhanced the formation of sulfate and nitrate, compared to that in the system without dust particles. However, the influence of GDD on the heterogeneous chemistry was much greater than that of ATD. Based on the model analysis, GDD enhanced the sulfate formation mainly *via* its high photoactivation capability. In the case of $NO_2$ oxidation, dust-phase nitrate formation is mainly regulated by the buffering capacity of dust. The measured buffering capacity of GDD was two times greater than that of ATD, and consequently, the maximum nitrate concentration with GDD was nearly two times higher than that with ATD. The model also highlights that in urban areas with high $NO_x$ concentrations, hygroscopic nitrate salts quickly form *via* titration of the carbonates in the dust phase, but in the presence of $SO_2$, the nitrate salts are gradually depleted by sulfate.





## 1 Introduction

Large quantities of mineral dust particles are frequently ejected into the atmosphere through wind action. Airborne mineral dust is a major contributor to atmospheric particulate matter with an estimated annual emission of 1000-3000 Tg yr$^{-1}$ (Textor et al., 2006;Tegen and Schepanski, 2009). Airborne dust is essential for radiation balance (Sokolik and Toon, 1996;Sokolik et al., 2001;Balkanski et al., 2007), cloud condensation nucleation (Krueger et al., 2003;Liu et al., 2008;Tang et al., 2016), oceanic metal-ion cycles (Jickells et al., 2005;Mahowald et al., 2005;Schulz et al., 2012) and visibility impairment (Kim et al., 2001;Camino et al., 2015).

The surface of mineral dust particles can act as an important sink for atmospheric trace gases, such as $O_3$, $NO_x$ (e.g., NO and $NO_2$) and $SO_2$, and can enhance the production of oxygenated compounds (e.g., nitrate and sulfate)(George et al., 2015). For example, 50% to 70% of the annual average total sulfate concentration is estimated to be formed by the heterogeneous oxidation of $SO_2$ in the vicinity of dust sources (Dentener et al., 1996;Usher et al., 2003a). $NO_x$ reportedly adsorbs on the surfaces of metal oxides and rapidly forms surface nitrite ($NO_2^-$) ions and eventually nitrate ions *via* the reaction of two nitrite ions or a nitrite ion with gas-phase $NO_2$ (Underwood et al., 2001). During a dust event (Beijing, China, on March 28, 2015), Wang et al. (2017) observed that the heterogeneous reactions on dust are the major production mechanisms for nitrate, 19 µg m$^{-3}$, and sulfate, 7 µg m$^{-3}$. Furthermore, the heterogeneous uptake of $O_3$ is catalytic on the surface of metal oxides and results in the destruction of $O_3$ by the formation of a surface-bound atomic oxygen and an oxygen molecule (Michel et al., 2002;Usher et al., 2003b).

Several recent studies have shown significant increases in sulfate and nitrate concentrations due to the heterogeneous photooxidation of $SO_2$ and $NO_x$ on mineral dust surfaces. For example, using a flow chamber, Dupart et al. (2014) observed that the $NO_2$ uptake rate of Arizona Test dust (ATD) particles was 4 times greater under UV-A irradiation than in the dark. A chamber study by Park and Jang (2016) also showed a significantly higher (10 times higher) $SO_2$ reactive uptake coefficient on ATD under UV light (a mixture of UV-A and UV-B light) than that obtained in the dark. In another chamber study, Park et al. (2017) reported that the increase in the $SO_2$ kinetic uptake coefficient of Gobi Desert dust (GDD) particles was higher than that observed for the ATD particles. Field observations by Ndour et al. (2009) and Dupart et al. (2012) showed that the uptake



coefficients of tracers (e.g., $NO_2$ and $SO_2$) on authentic dust particles increased under sunlight compare to those in the dark.

Despite numerous studies on the heterogeneous photooxidation of tracers, the mechanism behind the in-particle chemistry remains largely unknown. One challenge is modeling the

5 photocatalytic process of semi-conductive metal oxides (e.g., $TiO_2$ and $Fe_2O_3$) in dust particles. This photocatalytic process results in the formation of electron-hole pairs that can react with a water molecule or absorbed oxygen on the dust surface to form oxidant radicals (e.g., OH radical and $HO_2$ radical) and oxidize tracers on dust particles (Linsebigler et al., 1995;Hoffmann et al., 1995;Thompson and Yates, 2006;Cwiertny et al., 2008). Additionally, the hygroscopic property

of mineral dust, which is dynamic due to the atmospheric process associated with the dust buffering capacity and inorganic composition, complicates the dust model. For example, Tang et al. (2015) reported decreased hygroscopic properties due to the formation of calcium sulfate *via* the reaction of calcium carbonate with sulfuric acid. Some inorganic salts in dust, such as magnesium sulfate and calcium nitrate, are hydrophilic and can be hydrated in low humidity environments (Liu et al.,

2008;Beardsley et al., 2013;Abdelkader et al., 2017). The chemical properties of mineral dust can also be changed by carboxylic acids absorbed on dust particles, which further react with alkaline dust components (Mochizuki et al., 2016). Therefore, deriving a mathematical model to describe the hygroscopicity of dust particles is important for accurately processing both the multiphase partitioning of tracers and the in-particle chemistry under ambient conditions.

In our recent modeling work (Yu et al., 2017), the heterogeneous oxidation of $SO_2$ was simulated in the presence of ATD. However, ATD particles have chemical and physical properties that are different from those of ambient mineral dust particles. To simulate the heterogeneous chemistry of tracers under ambient conditions, a model should include different authentic dusts with various surface areas, hygroscopic properties, photocatalytic capacities, and buffering

abilities.

In this study, the Atmospheric Mineral Aerosol Reaction (AMAR) model highlights three aspects to accurately predict the heterogeneous photooxidation of $SO_2$ and $NO_x$: (1) the photocatalytic production of OH radicals; (2) the dynamic hygroscopicity of mineral dust; and (3) the buffering capacity determined by the dust compositions. For example, the kinetic mechanisms

for the photoactivation processes of different dust particles (ATD and GDD) were established



using the AMAR model based on laboratory data from the photodegradation of an impregnated dye (malachite green) on a dust filter sample. A mathematical model for dust particle hygroscopicity was also integrated into the model based on hygroscopicity data from Fourier transform infrared (FTIR) spectra of fresh and aged particles. The buffering capacity of dust

particles was parameterized in the model by measuring the nitrate that formed *via* the photooxidation of $NO_x$ in the presence of dust particles (ATD or GDD) using an indoor chamber with different humidities (20%, 55% and 80%). The resulting AMAR model was then evaluated against chamber data obtained under ambient conditions using a large outdoor smog reactor at the University of Florida Atmospheric Photochemical Outdoor Reactor (UF-APHOR).

**2 Experimental section and model description**

**2.1 Sample preparation and characterization**

The Gobi desert dust particles (GDD) were collected from the dust deposition region (Tsogt-Ovoo Soum in the Umnugovi Province, Mongolia) between March and May 2015. The collected sample was sieved to less than 20 µm. The Arizona test dust particles (ATD) are a

15 commercialized dust sample (size range: 0–3 µm) (Power Technology Inc. USA) from Arizona, USA. The particle size distributions of airborne dust particles were measured using both a scanning mobility particle sizer (SMPS; TSI 3080, USA) and an optical particle sizer (OPS; TSI 3330, USA). The measured SMPS and OPS data were merged using the Multi-Instrument Manager (MIM) 2.0 software (TSI, USA). An example of ATD and GDD particle distributions used in this

study is shown in Fig. S1. The concentration of geometric surface area ($cm^2$ $cm^{-3}$) of airborne dust particles were calculated based on the particle size distribution. The BET surface areas, which were measured using the BET method and a NOVA 2200 instrument, of ATD and GDD were previously reported to be 47.4 and 39.6 $m^2$ $g^{-1}$, respectively (Park et al., 2017).

**2.2 Indoor and outdoor chamber experiments**

The indoor and outdoor chamber operations have been previously reported (Yu et al., 2017;Park et al., 2017) (see Sect. S1). To generate the model parameters for the heterogeneous oxidation of $SO_2$, preexisting indoor chamber data were employed (Park and Jang, 2016). In this





study, nitrate data were added to create the model parameters for $NO_2$ oxidation. In the presence of different dust particles under various humidity levels (20%, 55% and 80%), $NO_2$ and $SO_2$ were photo-oxidized using a 2 $m^3$ indoor Teflon film chamber equipped with 16 UV lamps (wavelength range from 280 nm to 900 nm) (FS40T12/UVB, Solarc Systems Inc., Canada). The details on the

experimental conditions for the $NO_2$ oxidation are listed in Table S1. The resulting AMAR model was tested against the outdoor chamber data produced using the UF-APHOR dual chambers (52 + 52=104 $m^3$) under ambient environmental conditions. The nitrate and sulfate ion concentrations were measured using a particle into-liquid sampler (ADISO 2081, Applikon Inc., Netherlands) coupled with ion chromatography (761 Compact IC, Metrohm Inc., USA) (PILS-IC). The details

on the outdoor chamber data are listed in Table 1. The concentrations of $NO_x$, $SO_2$ and $O_3$ were continuously measured using a chemiluminescence NO/$NO_x$ analyzer (Model T201, Teledyne, USA), a fluorescence Total Reduced Sulfur (TRS) analyzer (Model 102E, Teledyne, USA) and a photometric ozone analyzer (Model 400E, Teledyne, USA), respectively.

## 2.3 Measurement of the dust particle photoactivation parameters

To parameterize the photoactivation capability of dust particles, a dust filter sample impregnated with a dye (malachite green) was photochemically irradiated using a specifically fabricated flow chamber equipped with a UV lamp (11SC-2.12; Pen-Ray., UK) coupled to a cut-off lens ($\leq 280 \pm 5$ nm wavelength, 20CGA-280; Newport, USA) (Fig. S2). The dry dust particles were introduced into the indoor chamber by passing clean air through a nebulizer (Pari LC star,

Starnberg, Germany). The dust particles were then collected on a Teflon-coated, glass-fiber filter (Emfab TX40 HI20 WW; Pallflex Corp., Putnam, CT) to obtain 200 µg of dust particles per filter. This filter sample was then impregnated with 4 µg of malachite green dye dissolved in ethanol. Afshar et al. (Afshar et al., 2011) reported that malachite green dye decays in the presence of metal oxides under UV light. The dye-impregnated dust filter sample was placed in a UV flow chamber

to activate the heterogeneous photodegradation of the dye on the dust particles. The humidity inside the flow chamber was controlled by manipulating the air flow (~0.5 L $min^{-1}$) and passing clean, dry air through a water bubbler. The degradation of the dye impregnated on the dust sample was then measured using a reflective UV-visible spectrophotometer (Jaz Spectrometer; Ocean





Optics Inc., USA). Figure S3 shows an example of measured light absorbance of dye impregnated dust filter before and after irradiation using UV light.

## 2.4 Hygroscopic properties of dust particles

The hygroscopic properties of the fresh and aged dust particles were determined using an FTIR spectrometer (Nicolet Magma 560, Madison, WI, USA) combined with a specifically fabricated optical flow chamber (Zhong and Jang, 2014;Jang et al., 2010;Beardsley et al., 2013;Park et al., 2017) that could control the humidity level in the range from 10% to 80%. The dust particles were impacted onto a silicon FTIR window (13×2 mm; Sigma–Aldrich, St. Louis, MO, USA) and weighed using an analytical balance (MX5; Mettler-Toledo Ltd., England). The

FTIR peak at 1650 cm$^{-1}$ was used to determine the water content of the particles. To calibrate the water content in the dust particles, $(NH_4)_2SO_4$ particles were used, and the calibration was based on the particle mass and water content estimated using an inorganic thermodynamic model (E-AIM II) (Clegg et al., 1998;Wexler and Clegg, 2002;Clegg and Wexler, 2011).

## 3 Results and discussion

**3.1 Description of the AMAR model**

     The AMAR model was developed to predict the heterogeneous oxidation of $SO_2$ and/or $NO_x$ in the presence of authentic mineral dust particles. As described in previous work (Yu et al., 2017), the formation of mass concentrations of sulfate ($[SO_4^{2-}]$, μg m$^{-3}$) and nitrate ($[NO_3^-]$, μg m$^{-3}$) is processed in three phases: the gas phase, inorganic salt-seeded aqueous phase and dust phase.

The key components of the model consist of multiphase tracer partitioning and the kinetic mechanisms of the three phases. Ambient dust particles are typically coated in multilayer water (Gustafsson et al., 2005;Ibrahim et al., 2018). Therefore, we assume that the gas–dust partitioning of tracers on multilayer water occurs via absorption. The partitioning coefficients of these gases can be calculated using Henry's Law constant ($K_H$), and the coefficients are influenced by the dust

phase water content. The oxidation of $SO_2$ and $NO_x$ in the gas phase and inorganic salt-seeded aqueous phase was simulated using the mechanisms previously reported in the literature (Liang and Jacobson, 1999;Binkowski and Roselle, 2003;Byun and Schere, 2006;Sarwar et al.,



2013;Sarwar et al., 2014;Yu et al., 2017). Dust-phase sulfuric acid partially or fully react with indigenous alkaline salt or the gaseous ammonia originating from the chamber wall (Li et al., 2015;Beardsley and Jang, 2016). For the inorganic salted aerosol (non-dust phase), neutralization is solely by gaseous ammonia. Inorganic salted aerosols were acidic (nearly ammonium bisulfate)

and they were not effloresced under our chamber experimental condition (Colberg et al., 2003). Therefore, heterogeneous chemistry in aqueous phase attributed to the oxidation of $SO_2$ and $NO_2$ during the entire chamber simulation.

An overall schematic of the dust-phase chemistry mechanism in the AMAR model is shown in Fig. 1 (also see Table S2). To accurately process the heterogeneous oxidation of $SO_2$ and

10 $NO_x$ under ambient conditions, we emphasized the three key processes in dust-phase chemistry:

(1) A mathematical model for dust particle hygroscopicity was derived to dynamically simulate the dust-phase water content as a function of dust aging, e.g., the neutralization of alkaline carbonates and inorganic components containing ammonia, sulfate and nitrate. This hygroscopic model improved the multiphase tracer partitioning and in-particle chemistry (Sect.

3.2).

(2) Kinetic mechanisms to simulate the photoactivation of dust particles and the formation of dust-phase OH radicals were included in the AMAR. Specifically, we standardized the technique to parameterize the photoactivation capability of various dust particles (Sect. 2.3 and Sect. 3.3).

(3) The neutralization mechanisms for dust particles with inorganic acids were systematically

approached using the buffering capacity parameter. This process is linked to the hygroscopicity of dust particles (Sect. 3.4).

## 3.2 Dust-phase water content

The inorganic salts and metal oxides in dust particles can absorb water *via* a thermodynamic equilibrium process and form a thin film of water on the dust surface. In general,

a higher water content enhances multiphase partitioning of tracers and the production of oxidized products (HONO, sulfate and nitrate). In the AMAR model, an equation for the dust-phase water content ($F_{water}$, $\mu g\ \mu g^{-1}$), which is defined as the water mass normalized by the dry dust mass, is mathematically derived. $F_{water}$ is estimated by an additive function with three parts:



$$F_{water} = a(e^{b \cdot RH} - 1) + c \cdot e^{d \cdot RH} \frac{[NO_3^-]}{[Dust]} + \frac{[Water]_{SO_4^{2-} - NH_4^+ - H_2O}}{[Dust]} \qquad (1)$$

where RH represents the relative humidity and ranges from 0 to 1. The first term, $a(e^{b \cdot RH} - 1)$, in Eq. (1) is associated with the water content of fresh dust particles. The 2$^{nd}$ term, $c \cdot e^{d \cdot RH} \frac{[NO_3^-]}{[Dust]}$, represents the hygroscopicity of the hydrophilic nitrate salts that are formed *via* titration of the

5 dust constituents (e.g., alkaline carbonates and some metal oxides). The 3$^{rd}$ term, $\frac{[Water]_{SO_4^{2-} - NH_4^+ - H_2O}}{[Dust]}$, originates from the ammonium sulfate system and is estimated *via* the inorganic thermodynamic model E-AIM II (Clegg et al., 1998;Wexler and Clegg, 2002;Clegg and Wexler, 2011). Coefficients a (0.03±0.01), b (3.6±0.5), c (1.4±0.4) and d (4.0±0.4) are dimensionless and they were determined using FTIR data (Fig. 2).

To determine the coefficients a and b, the hygroscopicity of fresh ATD particles or fresh GDD particles was measured using an FTIR spectrometer for RH levels from 10% to 80%. Similarly, the coefficients c and d were obtained from the FTIR spectra of aged dust particles, e.g., NO$_2$ photooxidation in the presence of ATD particles or GDD particles. The nitrate concentrations (µg µg$^{-1}$ in dust mass) were measured using PILS-IC and were 0.001 (approximately negligible)

for fresh ATD and 0.011 for aged ATD. The nitrate concentrations were 0.007 for fresh authentic GDD and 0.02 for aged GDD. Figure 2 shows $F_{water}$ values for ATD and GDD particles with and without aging. For both the fresh and aged dust particles, $F_{water}$ value gradually increases in the dry region (RH < 40%) but rapidly increases for RH values greater than 40%. $F_{water}$ of fresh GDD is higher than that of ATD for the entire RH range due to the presence of more hydrophilic nitrate

salts. Assuming that $F_{water}$ from the 2$^{nd}$ term has a linear relationship with the nitrate content, the $F_{water}$ value associated with nitrate salts can be estimated. Figure 2(b) shows that when the nitrate-associated $F_{water}$ is excluded, the $F_{water}$ value of fresh GDD (e.g., the hygroscopicity solely originating from dust constituents other than nitrates) is similar to that of ATD. The difference in model parameters for hygroscopicity between ATD and GDD is insignificant. Overall, clear phase

transitions and obvious differences between the hydration and dehydration processes were not observed for either types of dust particles. This trend suggests that the hygroscopicity of dust particles is caused by a variety of chemical species.



### 3.3 ATD and GDD photoactivation parameters

Mineral dust plays a key mechanistic role as a photocatalyst to accelerate tracer oxidation in the dust phase. The photoactivation of semiconducting metal oxides (M*) in dust particles can yield an electron-hole pair ($e^-_{cb}$-$h^+_{vb}$) that further reacts with water or oxygen molecules to form

5   oxidizing radicals, such as OH radicals (Linsebigler et al., 1995;Hoffmann et al., 1995;Thompson and Yates, 2006;Cwiertny et al., 2008;Yu et al., 2017).

$$M^* \xrightarrow{h\upsilon} M^* + e\_h \qquad\qquad k^j_{e\_h} \qquad\qquad (R1)$$

where e_h is an $e^-_{cb}$-$h^+_{vb}$ pair and $k^j_{e_h}$ is the operational photoactivation rate constant of dust particles. The production rate of the $e^-_{cb}$-$h^+_{vb}$ pair is described as

$$\frac{d[e\_h]_{dust}}{dt} = k^j_{e\_h}[M^*]_{dust} \qquad\qquad (2)$$

where $[M^*]_{dust}$ is the concentration (molecules cm$^{-2}$) of M* on the dust surface (calculate based on geometric surface area). In our recent study (Yu et al., 2017), $k^j_{e\_h}$ was linked to the wave-dependent mass-absorbance cross section and quantum yield of a given dust particle (ATD) (Fig. S4).  However, the type and quantity of conductive constituents in authentic dust particles vary.

Hence, to extend the model to ambient conditions, the photoactivation of different dust particles and their kinetic mechanisms must be estimated.

In this study, we determined the relative photoactivation rate constant for different dust particles using colorimetry integrated with a fabricated photochemical flow reactor (also see Sect. 2.3). The impregnated dye (malachite green) on the dust surface was photodegraded by the

oxidants created by the dust particles. The relative degradation rate constant of the dyed filter was measured using an online reflective UV-visible spectrometer to scale the photoactivation of the dust. The kinetic mechanisms for the reactions of the dye with radicals are expressed as follows

$$e\_h \rightarrow energy \qquad\qquad k_{recom} \qquad\qquad (R2)$$
$$e\_h + O_2 \rightarrow OH + O_2 \qquad\qquad k_{OH,O_2} \qquad\qquad (R3)$$
$$e\_h + H_2O \rightarrow OH + H_2O \qquad\qquad k_{OH,H_2O} \qquad\qquad (R4)$$
$$OH + dye \rightarrow dye' \qquad\qquad k_{dye} \qquad\qquad (R5)$$

where $k_{recom}$ is the rate constant of the recombination reaction of an electron with a hole. The concentration of the dye on the dust surface was assumed to be significantly higher than that of





the surface OH radical. The concentration unit (molecules cm$^{-2}$) of the chemical species in R1-R5 was multiplied by the geometric surface area concentration of the airborne dust particles ($A_{dust}$, cm$^2$ cm$^{-3}$) to convert to the concentration unit in air (molecules cm$^{-3}$). By combining R1-R5, the kinetic reaction rates for the e$^-_{cb}$-h$^+_{vb}$ pairs, OH radicals and dye can be written as follows.

$$5 \qquad \frac{d[e\_h]_{air}}{A_{dust}dt} = k^j_{e\_h}\frac{[M^*]_{air}}{A_{dust}} - k_{recom}\frac{[e\_h]_{air}}{A_{dust}} - k_{OH,O_2}\frac{[e\_h]_{air}[O_2]_{air}}{A^2_{dust}} - k_{OH,H_2O}\frac{[e\_h]_{air}[H_2O]_{air}}{A^2_{dust}} \qquad (3)$$

$$\frac{d[OH]_{air}}{A_{dust}dt} = k_{OH,O_2}\frac{[e\_h]_{air}[O_2]_{air}}{A^2_{dust}} + k_{OH,H_2O}\frac{[e\_h]_{air}[H_2O]_{air}}{A^2_{dust}} - k_{dye}\frac{[OH]_{air}[dye]_{air}}{A^2_{dust}} \qquad (4)$$

$$\frac{d[dye]_{air}}{A_{dust}dt} = -k_{dye}\frac{[dye]_{air}[OH]_{air}}{A^2_{dust}} \qquad (5)$$

The concentration of chemicals with the subscript "air" is the concentration in air (molecules cm$^{-3}$). Under the assumption of a steady state for the net reaction rate of an e$^-_{cb}$-h$^+_{vb}$ pair and OH radical, the dye consumption rate can be written as

$$\frac{d[dye]_{air}}{dt} = -\frac{k^j_{e\_h}[M^*]_{air}}{\frac{k_{recom}}{k_{OH,O_2}[O_2]_{dust}+k_{OH,H_2O}[H_2O]_{dust}} + 1} \qquad (6)$$

where $k_{recom}$ is much larger than $k_{OH,O_2}[O_2]_{dust}$ or $k_{OH,H_2O}[H_2O]_{dust}$ in Eq. (6). A previous study by Khorasani et al. (2014) also reported a recombination rate (~$10^4$ s$^{-1}$) for an e$^-_{cb}$-h$^+_{vb}$ pair on silicon that is much faster than the rate observed for typical in-particle reactions. Therefore, the term ($\frac{k_{recom}}{k_{OH,O_2}[O_2]_{dust}+k_{OH,H_2O}[H_2O]_{dust}}$) in Eq. (6) is much larger than 1. $[O_2]_{dust}$ is calculated through the partitioning process as follows

$$[O_2]_{dust} = K_p[O_2]_{gas}[H_2O]_{dust} \qquad (7)$$

where $[O_2]_{gas}$ is the concentration of oxygen in the air and $K_p$ is the partitioning coefficient for $O_2$ on the dust-phase water layer. By applying Eq. (7) to (6), the analytical solution for Eq. (6) can be written as

$$\Delta[dye]_{dust} = -k^j_{eh}\left(\frac{k_{OH,O_2}K_p[O_2]_{gas}+k_{OH,H_2O}}{k_{recom}}\right)[M^*]_{dust}[H_2O]_{dust}t \qquad (8)$$

As shown in Eq. (8), the dye decomposition on the particle surface is proportional to [M*], which changes based on the dust type, and the dust-phase water concentration, which can be estimated using $F_{water}$ and changes with the dust composition. Figure 3 shows the dye degradation rate in the presence of ATD or GDD particles, and the rate was measured using a UV flow chamber (Sect.





2.3). The [M*] value, which leverages the photoactivation ability of dust particles, is included in Fig. (3). The estimated [M*] for GDD is 2.55 times higher than that of ATD.

## 3.4 Impact of the dust buffering capacity

The buffering capacity is determined by the neutralization of the dust-phase constituents
(e.g., alkaline carbonates and some metal oxides) with inorganic acids. For example, alkaline carbonates in dust particles can react with nitric acid or sulfuric acid to form alkaline salts.

$$CaCO_3 + H_2SO_4 \rightarrow CaSO_4 + CO_2 \uparrow + H_2O \tag{R6}$$

$$CaCO_3 + HNO_3 \rightarrow Ca(NO_3)_2 + CO_2 \uparrow + H_2O \tag{R7}$$

In contrast to nitrate, sulfate can accumulate at levels beyond the neutralization capacity of dust
because sulfuric acid is not volatile in ambient humidity levels. Furthermore, sulfuric acid can deplete the nitrate salts that build up in the dust phase *via* the following reaction.

$$Ca(NO_3)_2 + H_2SO_4 \rightarrow CaSO_4 + 2HNO_3 \uparrow. \tag{R8}$$

The buffering capacity determines the maximum nitrate concentration that can build up on dust particles. Nitrate ions are hydrophilic and significantly influence the hygroscopicity of dust
particles ($F_{water}$ in Sect. 3.2). The buffering capacity was incorporated into the kinetic mechanisms in the AMAR model to dynamically modulate the $F_{water}$ value.

The buffering capacities of two different mineral dusts (ATD and GDD) were semi-empirically determined by fitting the nitrate prediction to the experimental data shown in Fig. S5 (experimental conditions in Table S1) using the kinetic mechanisms (R7 and gas-particle nitric
acid partitioning) in the AMAR model. The buffering capacity was determined using the maximum nitrate salt mass normalized by the dust mass (Sect. 2.1). The measured buffering capacities of ATD and GDD are 0.011 $\mu g \, \mu g^{-1}$ and 0.020 $\mu g \, \mu g^{-1}$, respectively.

## 3.5 Simulation of outdoor chamber data using the AMAR model

The resulting AMAR model was tested against the outdoor chamber data obtained from
25 simulating the oxidation of $NO_x$ (Fig. 4(a) and 4(a)) or $SO_2/ NO_x$ (Fig. 4(c) and 4(d)) in the presence of mineral dust particles under ambient sunlight. As shown in Fig. 4(a) and 4(b), nitrate rapidly formed in the morning, and the model well modeled the chamber data. Additionally, the nitrate mass normalized by the dust mass was higher for GDD than ATD. In addition, nitrate



depletion was observed (Fig. 4(a) and 4(b)) even in the absence of SO₂. The nitrate depletion in the chamber data is possibly due to the nitrate salts reacting with the carboxylic acids present in the chamber air, but the current model cannot predict this reaction. As shown in Fig. 4(c) and 4(d), the model well predicts the sulfate and nitrate concentrations produced from SO₂ oxidation at two

different NOₓ levels in the presence of GDD particles. The oxidation of SO₂ was suppressed when the NOₓ concentration was high because SO₂ competes with NO₂ to react with the OH radicals that form on dust surfaces. In the presence of SO₂, the model reasonably predicts the nitrate profile and shows that the nitrate quickly builds up in the morning and is moderately depleted by the formed sulfate. The of SO₂, NOₓ, ozone and dust particle concentrations are simulated in Fig. S6.

Figure 4 illustrates the predicted $F_{water}$ values with aging. The $F_{water}$ value is mainly influence by the humidity, which is high in the morning and gradually decreases as the temperature increases. However, $F_{water}$ is also modulated by the mineral dust particle aging process. For example, although the humidity level decreases between 8 AM and 10 AM, the $F_{water}$ value noticeably increases and coincides with the hygroscopic nitrate concentration time profile. The

$F_{water}$ value is significantly lower in the presence of SO₂ (Fig. 4(d)) than its absence (Fig. 4(b)) because the sulfate salts on dust particles and sulfates with ammonium ions (e.g., more titrated than ammonium hydrogen sulfate) are less hygroscopic than nitrate salts.

### 3.6 Model sensitivity

The sensitivity of the model predictions for nitrate (Fig. (5)) and sulfate (Fig. (6)) to the

major input variables (e.g., relative humidity, temperature, sunlight intensity, dust mass concentration and NOₓ concentration) was evaluated. The sensitivity test was mainly performed for GDD particles (100 µg m⁻³) under the environmental conditions at Gainesville, Florida on 23 November 2017. The nitrate and sulfate mass concentrations in Fig. 5 and 6, respectively, are normalized with the dust mass.

As shown in Fig. 5(a) and 6(a), the formation of both nitrate and sulfate is significantly sensitive to the RH level, but the reasons for this sensitivity are different. There was a sudden increase in the nitrate concentration between a low RH (20% and 55%) and a high RH (80%), imitating the $F_{water}$ trend. In addition to the nitrate salt formation, which is influenced by the buffering capacity, the partitioning of hydrophilic nitric acid into the water layer increases at a



higher RH. Unlike nitrate, the sulfate concentration gradually increases as the RH increases. The $F_{water}$ value of the sulfate salts (Fig. 6(a)) is relatively smaller than that of the nitrate salts (Fig. 5(a)). Additionally, nitrate formation is more sensitive to temperature than sulfate formation due to the nitric acid partitioning process. For the different dust types (ATD vs. GDD), the formation

of nitrate (Fig. 5(c)) and sulfate (Fig. 6(c)) is higher with GDD. As discussed in Sect. 3.4, the maximum amount of nitrate salts in the dust phase is determined by the buffering capacity of the dust particles. The buffering capacity of GDD is two times higher than that of ATD (e.g., 0.011 µg µg$^{-1}$ for ATD and 0.020 µg µg$^{-1}$ for GDD), and thus, the nitrate concentration in the GDD system is nearly two times higher than that in the ATD system. Another reason for the high sulfate

formation in the presence of GDD is the photoactivation ability of GDD. An in-depth explanation will be presented in Sect. 4. The sunlight intensity has more of an impact on sulfate formation than nitrate formation, as seen in Fig. 5(d) and Fig. 6(d). Although nitrate formation is accelerated by strong sunlight, the nitrate production under the different sunlight intensities is governed by the buffering capacity of a given dust type (e.g., GDD).

Figure 6(e) shows the sulfate formation sensitivity to three levels of NO$_x$ (2, 20 and 40 ppb NO$_x$) under ambient conditions (e.g., sunlight, temperature and humidity). In the presence of NO$_x$ (40 ppb), the sulfate formation sensitivity to three different RH levels (20, 55, and 80%) was tested, as shown in Fig. 6(f). In general, sulfate is suppressed by increasing NO$_x$ concentrations (Fig. 6(e)). Similar to the effects of humidity on nitrate production at a low NO$_x$ level (Fig. 6(a)), the nitrate

formation with a higher NO$_x$ concentration (40 ppb) is also enhanced by a higher RH level, as seen in Fig. 6(f). Additionally, Fig. 6(g) shows how the total sulfate can be attributed to sulfate originating from the reactions in the different phases: (1) the sulfate from the gas phase and inorganic salt-seeded aqueous phase and (2) the sulfate from the dust phase. Dust-phase sulfate formation is suppressed by NO$_x$ due to competition between the absorbed SO$_2$ and NO$_2$ for surface

OH radicals, while sulfate formation in the inorganic salt-seeded aqueous phase is promoted by NO$_x$. When the %RH increases from 20 to 80, the heterogeneous reaction is significantly promoted due to the large $F_{water}$ value that enhances both the partitioning process and the production of OH radicals on dust surfaces.





## 4 Model uncertainties

To characterize the impact of dust characteristics on sulfate formation, the heterogeneous oxidation of $SO_2$ in the presence of five different dust types, including ATD, GDD, and three artificially formulated dusts (Dust I, II and III), was compared. As shown in Fig. 7, the three
characteristic parameters of the dust particles, including the photoactivation capability ($[M^*]_{dust}$ in Eq. (2)), buffering capacity (Sect. 3.4), and hygroscopicity ($F_{water}$ in Sect. 3.2), were scaled relative to the ATD particles. The relative values of the three parameters for GDD were obtained using laboratory data. The simulation with three artificially formulated samples, Dust I, II, and III, was used to analyze why GDD particles have a larger influence on sulfate formation than ATD and
which dust characteristic parameters are the most important for sulfate formation. Figure 7 illustrates that the photoactivation ability of dust ($[M^*]_{dust}$) is the most important among the three parameters. For example, the sulfate formation noticeably increased between ATD and Dust I. When the three characteristic dust parameters are determined by laboratory studies, in the future, the model can simulate the impact of authentic dust particles on sulfate formation.

Figure S7 also shows the uncertainty in the sulfate and nitrate predictions in the presence of GDD using the AMAR model based on three major dust characteristic parameters (e.g., $F_{water}$, buffering capacity and photoactivation capability). Assuming that the sulfuric acid beyond the buffering capacity of GDD is treated by the $NH_4^+$-$SO_4^{2-}$-$H_2O$ system, we estimated the $F_{water}$ value using an inorganic thermodynamic model with a large uncertainty (E-AIM II) (Clegg et al.,
1998;Wexler and Clegg, 2002). In the model simulation, the ±10% uncertainty in $F_{water}$ result in ±7.8% variation in the sulfate concentration and -0.9% to 1.2% variation in the nitrate concentration. As shown in Fig. 3, the uncertainty in the photoactivation parameters of dust particles varies with the RH; e.g., the uncertainty is higher at higher RH levels. The probable uncertainty for the photoactivation of GDD particles at a high %RH (80%) is ±50% and results in
25 -47.7% to 55.7% variation in the sulfate concentrations and -1.0% to 1.9% in the nitrate concentration. The uncertainty in the buffering capacity (±10%) is associated with using ion chromatography to measure the ion concentrations and yields -0.7% to 0.8% variation in the sulfate concentration and -7.6% to 9.4% in the nitrate concentration.





## 5 Atmospheric implications

Dust storms originating from the Gobi Desert often outbreak during the spring season and influence the air quality over polluted urbans or industrial areas in East Asia (Hsu et al., 2010;Li et al., 2012). In a typical, polluted urban environment; e.g., where $NO_x$ and $SO_2$ levels are high (40

5 ppb of $NO_x$, 5 ppb of $SO_2$ and 200 µg m$^{-3}$ of GDD), the AMAR model shows authentic dust particles are quickly saturated with nitrate and sulfate (concentrations higher than the buffering capacity of GDD), as shown in Fig. S8(a). Under the high $NO_x$ conditions in most urban areas, the heterogeneously formed nitrate on the dust particles modulates the dust hygroscopicity, which is generally higher than that of fresh dust particles. Under high $SO_2$ concentrations (e.g., 20 ppb of

10 $SO_2$, Fig. S8(b)), the dust-phase sulfate depletes nitrate, as discussed in Sect. 3.4. Therefore, we conclude that $SO_2$ and $NO_x$ rapidly convert into the nitrate or sulfate concentrations during dust break episodes.

Under ambient conditions, the photooxidation of hydrocarbons in the presence of $NO_x$ is indispensable for the formation of ozone. In the model, the absorbed ozone on dust surfaces

positively modulates the formation of sulfate and nitrate *via* either the autoxidation mechanism or the production of OH radicals (Yu et al., 2017). Although $NO_2$ generally suppresses the formation of sulfate, its influence on heterogeneous chemistry of $SO_2$ is compounded with ozone in ambient air. For example, heterogeneous chemistry of ozone becomes important in nighttime, particularly when humidity is high, and promotes $SO_2$ oxidation. Additionally, some organic compounds can

sink onto dust surfaces *via* a partitioning process and complicate the heterogeneous chemistry in the model. For example, the organic carboxylic acids on dust surfaces can react with alkaline carbonates to form alkaline carboxylates. Beardsley et al. (2013) reported that anions in inorganic aerosols, such as $NO_3^-$, can be depleted by the formation of carboxylic acids, and the subsequent change aerosol hygroscopic properties. Semivolatile organic compounds compete with the

absorbed $SO_2$ and $NO_2$ for the consumption of OH radicals. Therefore, the model requires further in-depth dust chemistry of organic compounds in the future to accurately predict sulfate and nitrate formation in ambient environments.



**Acknowledgments**

This research was supported by the award from the Ministry of Science and ICT, the Ministry of Environment, the Ministry of Health and Welfare (2017M3D8A1090654), the award from the National Institute of Metrological Sciences (KMA2018-00512), and the Scholarship from
5 the Fulbright Scholar (from USA to Mongolia).



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



**Table 1.** Outdoor chamber experimental conditions for $NO_x$ and $SO_2$ oxidation in the presence of Gobi Desert dust (GDD) particles and Arizona Test dust (ATD) particles.

| Exp. No | Purpose | Type of particles | Mass conc. of particles[a] ($\mu g\ m^{-3}$)[b] | RH[b] (%) | Temp[b]. (°C) | Initial $NO/NO_2$ conc. (ppb)[b] | Initial $SO_2$ conc. (ppb)[b] | Initial $O_3$ conc. (ppb)[b] |
|---|---|---|---|---|---|---|---|---|
| 10/6/2017 | High and low $NO_x$ with $SO_2$ | GDD | 337.3 | 13.9-91.8 | 293.9-319.3 | 22.1/123.1 | 93.9 | 4.1 |
| | | GDD | 375.3 | 21.9-95.6 | 294.3-320.3 | 6.1/37.1 | 98.2 | 6.0 |
| 17/9/2017 | GDD vs. ATD with NOx | ATD | 334.0 | 14.2-50.9 | 293.6-319.4 | 19.1/108.1 | N.A. [c] | 3.6 |
| | | GDD | 408.1 | 21.0-61.6 | 294.0-318.9 | 17.1/99.1 | N.A. [c] | 2.8 |

[a] The mass concentrations of GDD and ATD particles were calculated from the SMPS data combined with OPC data. The density of dust particles is 2.65 g cm$^{-3}$ and the particle size distribution was calculated up to 3 µm.

[b] The errors associated with NO, $NO_2$, and $O_3$ were ±12.5%, ±6.9%, and ±0.2%, respectively. The error associated with dust mass were ±6% based on SMPS and OPC data. The accuracy of the measurement of RH and temperature were ±5 % and ±0.5 K, respectively.

[c] N.A.: not applicable (no $SO_2$ injection).

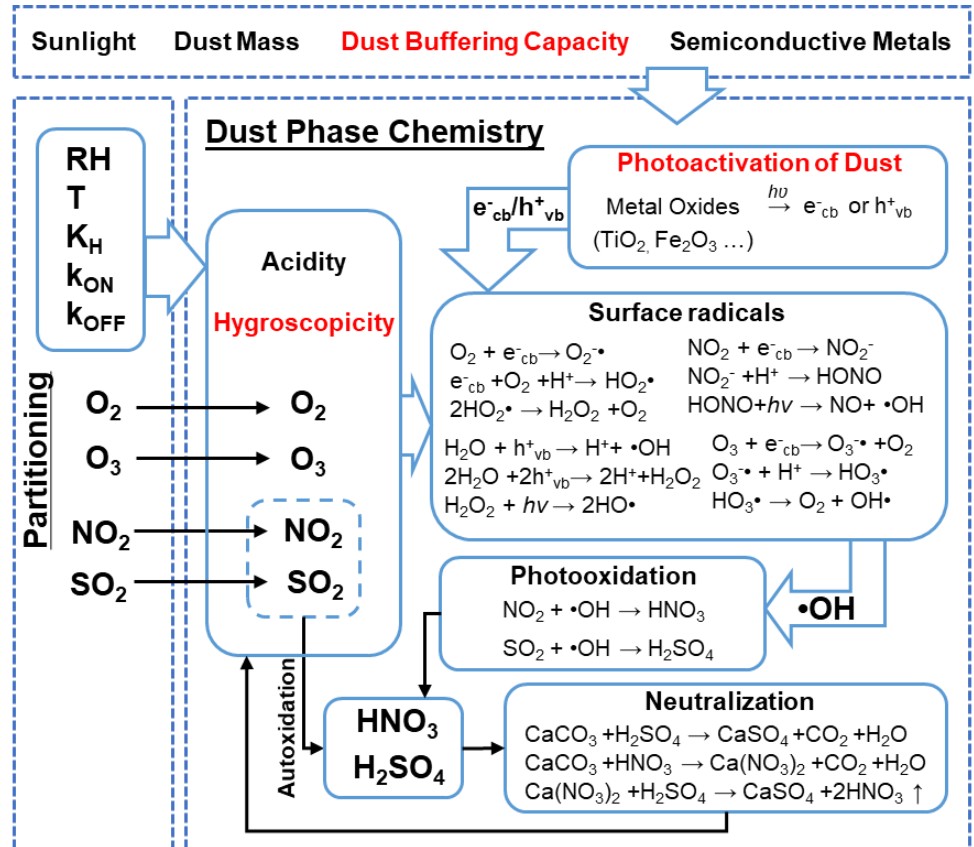

Figure 1. The overall schematic of dust phase chemistry in the AMAR model.





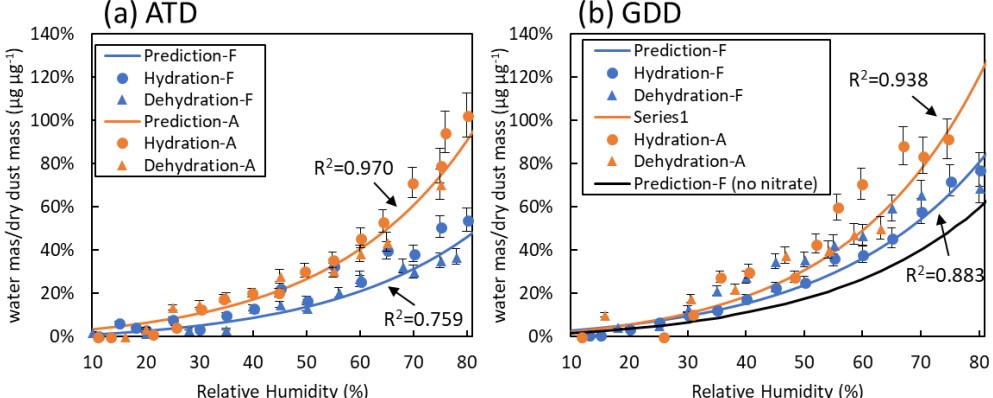

Figure 2. The fraction of water mass relative to the dry dust mass for fresh and photochemically aged (a) ATD and (b) GDD particles as a function of the relative humidity from 10% to 80%. The amount of water in a hydrate form (water content under RH <10%) was subtracted from the water content measured using FTIR (circle and triangle). "F" and "A" represent the fresh and aged dust particles, respectively. The water content for fresh GDD with theoretically no indigenous nitrate is also predicted and shown. The aged dust samples were collected from dust particles that were photochemically aged in the presence of $NO_x$. The estimated nitrate concentrations for fresh and aged ATD are 0.001 $\mu g \, \mu g^{-1}$ and 0.011 $\mu g \, \mu g^{-1}$, respectively. The estimated nitrate concentrations for fresh and aged GDD are 0.007 $\mu g \, \mu g^{-1}$ and 0.02 $\mu g \, \mu g^{-1}$, respectively. The error bars were estimated from the uncertainties in the FTIR absorbance measurements of the O-H band.





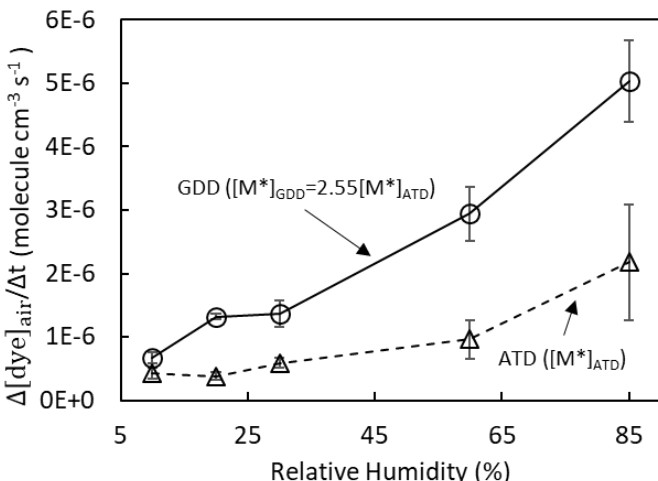

Figure 3. The dye degradation rate in the presence of ATD or GDD particles measured using a UV flow chamber under RH levels ranging from 10% to 85%. As a control, the photodegradation of malachite green in the absence of dust was measured, but the degradation was negligible. The error bars represent the standard deviations.





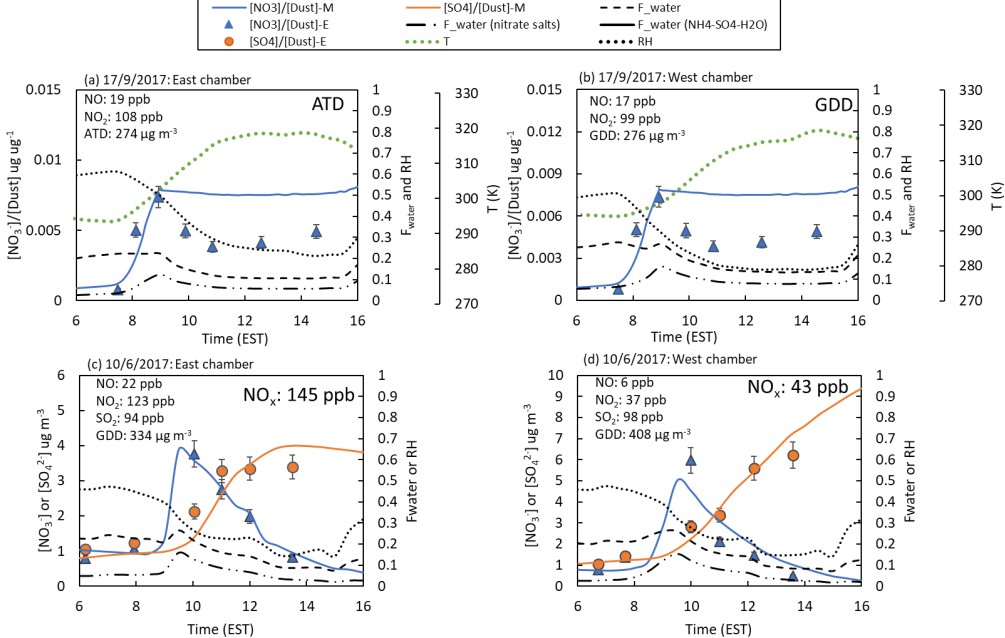

Figure 4. Simulation of the outdoor chamber data using the AMAR model for (A) ATD (2017-09-17) and (B) GDD (2017-09-17) particles in the presence of $NO_x$ and $SO_2$ oxidation on GDD particles in the presence of (C) high $NO_x$ (2017-06-10) and (D) low $NO_x$ (2017-06-10) concentrations. $F_{water}$(nitrate salts) and $F_{water}$($NH_4^+$-$SO_4^{2-}$-$H_2O$) are the second and third terms in Eq. (1) and represent the additional absorbed water by alkaline nitrate salts and the ammonium sulfate system, respectively. The simulation result was not correct for particle loss.





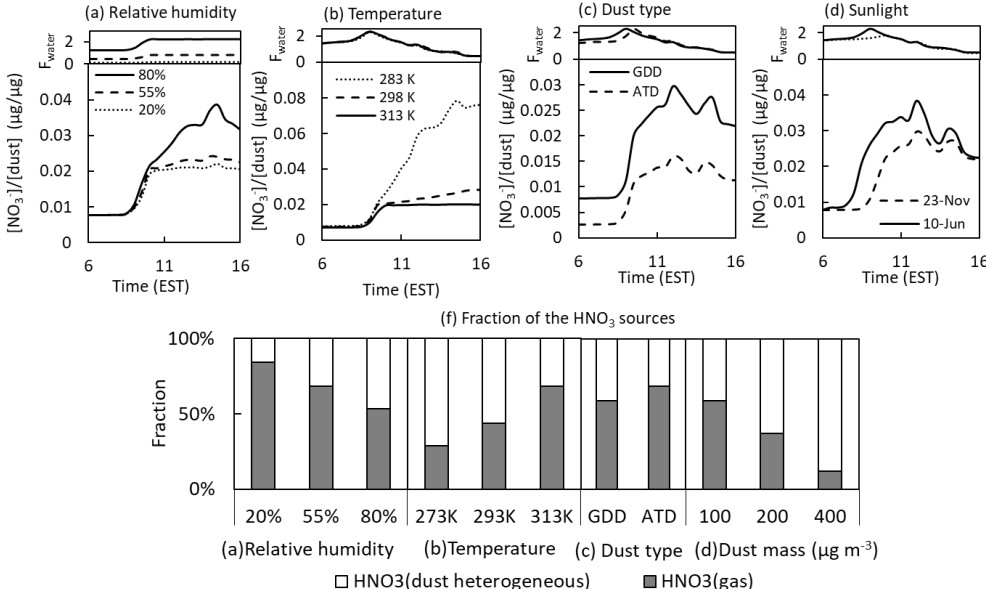

Figure 5. Sensitivity of the model nitrate prediction to the (a) relative humidity at 20%, 55% and 80%; (b) temperature at 273 K, 293 K and 313 K; (c) ATD vs. GDD particles; (d) concentration of GDD at 100, 200 and 400 µg m$^{-3}$; and (e) sunlight profile on 23 November 2017 vs. 10 June 2017. The fraction of the HNO$_3$ sources formed from the gas-phase reaction and dust-phase heterogeneous reaction to the total HNO$_3$ is shown in (f). The simulation was conducted with 100 µg m$^{-3}$ of initial GDD particles, 40 ppb of initial NO$_x$ (NO:NO$_2$=1:1), 2 ppb of initial O$_3$ and 10 ppb isoprene under ambient environmental conditions on 23 November 2017. The simulation was performed without considering particle loss.





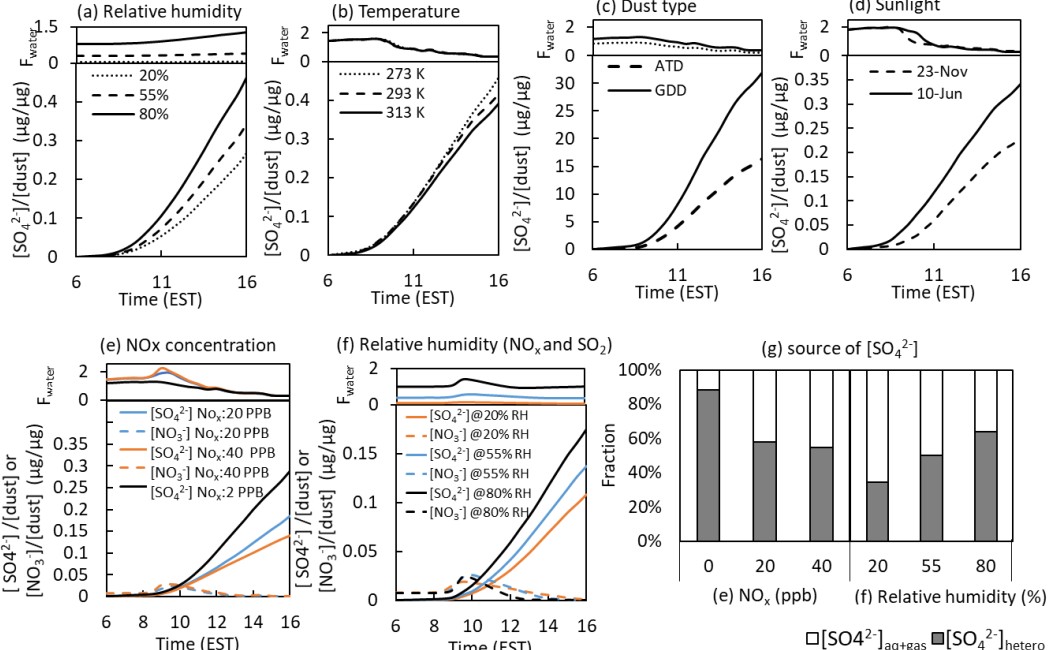

Figure 6. Sensitivity of the model predicted sulfate concentration to the (a) relative humidity at 20%, 55% and 80%; (b) temperature at 273 K, 293 K and 313 K; (c) sunlight profile (23 November 2017 vs. 10 June 2017); (d) dust type (ATD vs. GDD); (e) initial concentration of $NO_x$ (0, 20 and 40 ppb); and (f) relative humidity (20%, 55% and 80%) in the presence of 20 ppb of $NO_x$. The fraction of sulfate from the gas phase and non-dust aqueous phase ($[SO_4^{2-}]_{aq+gas}$) and the heterogeneously formed sulfate in the dust phase ($[SO_4^{2-}]_{hetero}$) relative to the total sulfate is shown in (g). The dust-phase nitrate and water content were also predicted. For the sensitivity test, the simulation was conducted with 100 µg m$^{-3}$ of initial GDD particles, 40 ppb of initial $SO_2$, 2 ppb of initial $O_3$ and 10 ppb isoprene under ambient environmental conditions on 23 November 2017. The simulation was performed without considering particle loss.



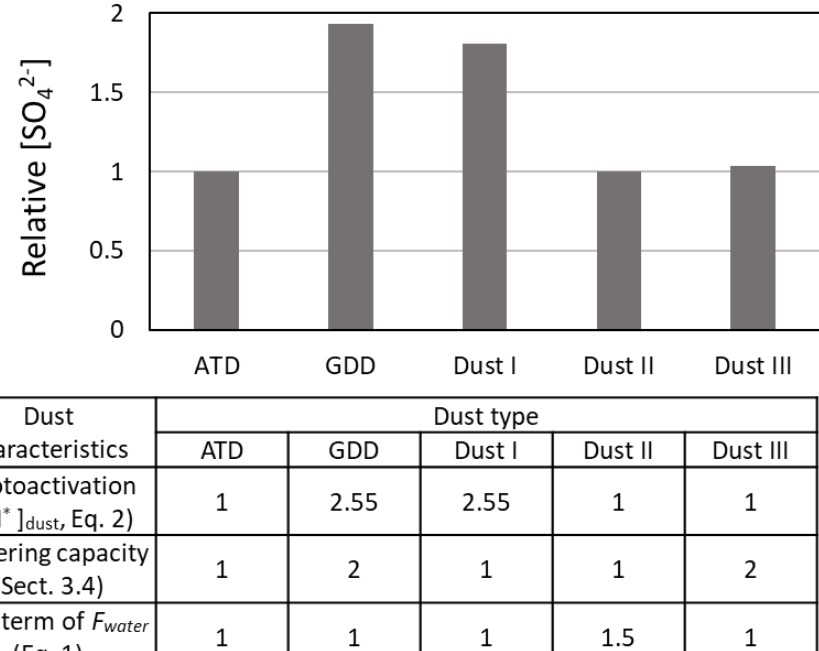

| Dust characteristics | Dust type | | | | |
|---|---|---|---|---|---|
| | ATD | GDD | Dust I | Dust II | Dust III |
| Photoactivation ($[M^*]_{dust}$, Eq. 2) | 1 | 2.55 | 2.55 | 1 | 1 |
| Buffering capacity (Sect. 3.4) | 1 | 2 | 1 | 1 | 2 |
| First term of $F_{water}$ (Eq. 1) | 1 | 1 | 1 | 1.5 | 1 |

**Figure 7**. The analysis of the influential parameters associated with dust characteristics to form sulfate. The relative concentration of sulfate is predicted using AMAR in the presence of different types of dust including ATD, GDD and three types of artificial dust (Dust I, II and III). The variation of dust type is determined *via* three major aspects: photoactivation capability of dust linked to [M*] in Eq. 2 (Sect. 3.3), the buffering capacity of dust (Sect. 3.4) and $F_{water}$ in Eq. 1 (Sect. 3.2). Dust I, II and III are artificially formulated to analyse how the three dust properties can influence the sulfate formation. ATD is used as a reference dust. The three parameters of GDD, which were obtained from experimental data, are scaled to those of ATD. For analysis, the simulation is conducted with 100 µg m$^{-3}$ of initial dust particles, 40 ppb of initial $SO_2$, 2 ppb of initial $O_3$ and 10 ppb isoprene under ambient environmental condition on 23 November 2017. The simulation was performed without considering the particle loss to the chamber wall.