# Peer review of "Simulation of Heterogeneous Photooxidation of SO2 and NOx in the presence of Gobi Desert Dust Particles under Ambient Sunlight"

_Atmospheric Chemistry and Physics, 2018_

## Referee Comment (RC1) · M. J. Tang (Referee) · 7 Jul 2018

Heterogeneous reactions of $SO_2$ and $NO_2$ with mineral dust affects the formation of nitrate and sulfate, and also impacts physicochemical properties of aerosol particles. Despite a number of studies carried out in the last 20 years, kinetics parameters have not been well constrained yet, especially under illuminated conditions. Yu and Jang carried out systematical laboratory work using an outdoor chamber, and developed a numerical model to describe these processes. The laboratory and modeling work is well done, and the manuscript is well written. I would like to recommend it for final publication after the following comments are adequately addressed.

**Scientifical comments:**

Page 11, line 1-2: why does DGG show higher photo-activation ability than ATD? Can this be explained by measured mineralogical components for these two types of dust?

Page 11, line 21-22: Why does GDD have higher buffering capacity than ATD? Is it related their carbonate contents? I would suggest the authors measure the carbonate (and iron oxides) contents for these two types of mineral dust.

**Technical comments:**

Uptake coefficients have been widely used to describe the rates of heterogeneous reactions of mineral dust. Can the author derive uptake coefficients for their experiments under different conditions and then compared these values with those reported in previous studies?

Page 2, line 11: Two important reviews papers on heterogeneous chemistry of mineral dust (Crowley et al., 2010; Tang et al., 2017) should be cited here.

Page 3, line 1 and line 8 (as well as a few other places in the manuscript): please change "tracers" to "trace gases".

**References:**

Crowley, J. N., Ammann, M., Cox, R. A., Hynes, R. G., Jenkin, M. E., Mellouki, A., Rossi, M. J., Troe, J., and Wallington, T. J.: Evaluated Kinetic and Photochemical Data for Atmospheric Chemistry: Volume V - Heterogeneous Reactions on Solid Substrates, Atmos. Chem. Phys., 10, 9059-9223, 2010.

Tang, M. J., Huang, X., Lu, K. D., Ge, M. F., Li, Y. J., Cheng, P., Zhu, T., Ding, A. J., Zhang, Y. H., Gligorovski, S., Song, W., Ding, X., Bi, X. H., and Wang, X. M.: Heterogeneous

reactions of mineral dust aerosol: implications for tropospheric oxidation capacity, Atmos. Chem. Phys., 17, 11727-11777, 2017.

---

## Referee Comment (RC2) · Anonymous Referee #1 · 9 Jul 2018

This is an interesting study where the SO2 and NOx oxidation on mineral dust was investigated by means of simulation chambers, and simulated using the Atmospheric Mineral Aerosol Reaction (AMAR) model. Different dust particles (Gobi desert GDD and Arizona test dust ATD) were considered and their differences in reactivity and buffer capacity are discussed. Overall this paper is well written and addresses an important topic (mineral dust is an important category of aerosols), I would therefore recommend its publication once the authors have had a chance to discuss the comments (some are major) raised below.

[Figure]

I have a conceptual problem with the AMAR model, which takes into account processes in three phases: the gas phase, inorganic salt-seeded aqueous phase and dust phase, implying that uptake is treated as absorption (according to Henry's law). However is many studies involving the uptake of traces gases on mineral dust, a Langmuir type behavior has been reported, showing an adsorption behavior that could typically contradict the assumption of absorption. Also, water and several gases have been show to exhibit competitive adsorption properties, going against the absorption assumption. Could it not be that this two assumption would correspond to two completely different humidity regimes? Maybe the authors could comment on that, and strengthen their assumptions in the manuscript.

Concerning the determination of the photoactivation parameters, have you checked how much is simply due to bleaching of the dye? That is a commonly reported issue in photocatalytic degradation of dyes on TiO2. In addition, the organic compound may also directly react with the electron-hole pair changing the reaction mechanism given in (R2) to (R5). Do you have any indication that this is occurring with the selected dye? By the way, did you performed any elemental analysis of the two samples, or just for the GDD as ATD has a known composition? This should then provide a basis for explaining the difference in the photoactivation parameters.

Sulfate is very often considered as a poison for surfaces, as it passivates very rapidly reactive surfaces. However, the outcome of the AMR does show that (probably due to the absorption assumption discussed above). Have you built in some capacity to have surface saturation or not at all?

Figure 4 seems to show that the model does capture the nitrate formation at longer times. How do you explain this? Is there any renoxification process taking place in this system?

I would recommend to the authors to review their paper and clarify a few basic assumptions in their work.

---

## Author Comment (AC1) · 26 Jul 2018

**Response to Reviewers' comments RC1 (Manuscript Ref. NO.: acp-2018-68)**

We appreciate the referees for the thoughtful and constructive comments on this manuscript. We believe that the quality of the manuscript has been greatly improved due to their valuable comments. The detailed responses to the specific questions from the reviewers were presented in the following.

Referee #1:

**Overall Comment for Referee #1:**

Heterogeneous reactions of $SO_2$ and $NO_2$ with mineral dust affects the formation of nitrate and sulfate, and also impacts physicochemical properties of aerosol particles. Despite a number of studies carried out in the last 20 years, kinetics parameters have not been well constrained yet, especially under illuminated conditions. Yu and Jang carried out systematical laboratory work using an outdoor chamber, and developed a numerical model to describe these processes. The laboratory and modeling work is well done, and the manuscript is well written. I would like to recommend it for final publication after the following comments are adequately addressed.

**Comment 1:** Page 11, line1-2: why does GDD show higher photo-activation ability than ATD? Can this be explained by measured mineralogical components for these two types of dust?

**Response:** In order to respond to the reviewer's comment, we added some explanation to the end of section 3.3. "This difference in dust's photoactivation ability can be explained by the dissimilarity in their elemental compositions. As seen in the previous study by (Park et al., 2017), the elemental fraction of conductive metals such as iron and tritium appeared to be higher with the GDD of this study than reference ATD. The correlation between the metal compositions and photoactivation ability of dust particles needs to be explored in future."

**Comment 2:** Page 11, line 21-22: Why does GDD have higher buffering capacity than ATD? Is it related their carbonate contents? I would suggest the authors measure the carbonate (and iron oxides) contents for these two types of mineral dust.

**Response:** We added the explanation that why GDD has the higher buffering capacity than ATD (at the end of Sect. 3.4: Impact of the dust buffering capacity). This explanation reads now, "The difference in buffering capacity between GDD and ATD originates from the content of alkaline carbonates and partially metal oxides. The element analysis measured by (Park et al., 2017) showed that GDD contained the greater amount of alkaline metals (e.g., K, Ca, Na and Mg) and transition metals (e.g., Fe and Ti) than ATD. However, the reaction generally occurs on the surface of dust rather than the whole body of dust due to its solidity and tortuosity. Thus, the actual buffering capacity of dust is much smaller than the total amount of alkaline carbonates and metal oxides in bulk dust."

**Comment 3:** Uptake coefficients have been widely used to describe the rates of heterogeneous reactions of mineral dust. Can the author derive uptake coefficients for their experiments under different conditions and then compared these values with those reported in previous studies?

**Response:** In order to respond to the reviewer, Sect. S2 was newly added into the revised supporting information and reads now.

The reactions of trace gases on the dust particles are traditionally expressed based on the first order reaction using the reactive uptake coefficient ($\gamma$). In AMAR model, the oxidation of trace gases in dust phase includes the 1st order and the 2nd order reactions (Table S2). Furthermore, the rate constants of heterogeneous reactions are photocatalytically and dynamically changing through day and night. For convenience, we calculate $\gamma$ of $SO_2$ and $NO_2$ using the gas-dust partitioning coefficients and the rate constants as follows (Yu et al., 2017),

$$\gamma_{dark,SO_2} = \frac{4K_{d,SO_2}k_{auto,SO_2}}{\omega_{SO_2}} \qquad \text{for } SO_2 \text{ autoxidation} \qquad (S1)$$

$$\gamma_{light,SO_2} = \frac{4K_{d,SO_2}\left(k_{photo,SO_2}[\text{OH(d)}]+k_{auto,SO_2}\right)}{\omega_{SO_2}} \qquad \text{for } SO_2 \text{ photooxidation} \qquad (S2)$$

$$\gamma_{dark,NO_2} = \frac{4K_{d,NO_2}k_{auto,NO_2}}{\omega_{NO_2}} \qquad \text{for } NO_2 \text{ autoxidation} \qquad (S3)$$

$$\gamma_{light,NO_2} = \frac{4K_{d,NO_2}\left(k_{photo,NO_2}[\text{OH(d)}]+k_{auto,NO_2}\right)}{\omega_{NO_2}} \qquad \text{for } NO_2 \text{ photooxidation} \qquad (S4)$$

$\omega$ (m s$^{-1}$) is the mean molecular velocity of gas species. $k_{auto}$ (s$^{-1}$) is the first order rate constant for autoxidation of SO$_2$ or NO$_2$ and $k_{photo}$ (cm$^3$ molecule$^{-1}$ s$^{-1}$) is the second order rate constant for photooxidation of SO$_2$ or NO$_2$ by OH radicals on dust particles. [OH(d)] (molecule per cc of air) is the concentration of OH radicals on dust. $K_d$ (m$^3$ m$^{-2}$) is the gas-dust partitioning coefficient and is calculated using the geometric surface concentration of airborne dust particles ($A_{dust}$, m$^2$ m$^{-3}$). $K_d$ can be calculated as,

$$K_d = \frac{[gas(d)]}{[gas(g)]A_{dust}} \text{ and} \tag{S5}$$

$$K_d = \frac{k_{up}}{k_{off}}, \tag{S6}$$

where [gas(d)] and [gas(g)] are the concentration of gas species in dust and gas phase, respectively. $k_{up}$ and $k_{off}$ is first calculated for SO$_2$ and then scaled using Henry's law constant for other gaseous compounds (Yu et al., 2017). Figure S4 illustrates the time profile of $\gamma$ under the ambient environmental conditions on November 23, 2017.

(a)

[Figure]

(b)

[Figure]

Figure S4. (a) Time profile of reactive uptake coefficient ($\gamma$) of $SO_2$ and $NO_2$ on Gobi Desert Dust particles under ambient sunlight. $\gamma$ is calculated using simulation results that are conducted with 200 μg m$^{-3}$ GDD particles, 40 ppb $SO_2$ and 20 ppb $NO_2$ under ambient conditions on 23 November 2017. The particle loss is not considered in the simulation. (b) Time profile of temperature (C°), relative humidity (%) and Total UV radiation (TUVR, W m$^{-2}$) on November 23, 2017 at Gainesville, Florida (latitude/longitude: 29.64185°/–82.347883°).

**Comment 4:** Page 2, line 11: Two important review papers on heterogeneous chemistry of mineral dust (Crowley et al., 2010; Tang et al., 2017) should be cited here.

**Response:** These two review papers have been cited in the revised manuscript.

**Comment 5:** Page 3, line 1 and line 8 (as well as a few other places in the manuscript): please change "tracers" to "trace gases".

**Response:** Word "tracers" has been changed to "trace gases" in the revised manuscript.

**Reference:**

Park, J., Jang, M., and Yu, Z.: Heterogeneous Photo-oxidation of $SO_2$ in the Presence of Two Different Mineral Dust Particles: Gobi and Arizona Dust, Environ Sci Technol, 51, 9605-9613, 10.1021/acs.est.7b00588, 2017.

Yu, Z., Jang, M., and Park, J.: Modeling atmospheric mineral aerosol chemistry to predict heterogeneous photooxidation of $SO_2$, Atmos Chem Phys, 17, 10001-10017, 2017.

---

## Author Comment (AC2) · 26 Jul 2018

**Response to Reviewers' comments RC2 (Manuscript Ref. NO.: acp-2018-68)**

We would like to thank the referee for the thoughtful comments on our work. We have carefully studied these comments and modified the manuscript. The detailed responses to the specific questions were presented in the following.

Referee #2:

**Overall Comment for Referee #2:**

This is an interesting study where the $SO_2$ and $NO_x$ oxidation on mineral dust was investigated by means of simulation chambers, and simulated using the Atmospheric Mineral Aerosol Reaction (AMAR) model. Different dust particles (Gobi desert GDD and Arizona test dust ATD) were considered and their differences in reactivity and buffer capacity are discussed. Overall this paper is well written and addresses an important topic (mineral dust is an important category of aerosols), I would therefore recommend its publication once the authors have had a chance to discuss the comments (some are major) raised below.

**Comment 1:** I have a conceptual problem with the AMAR model, which takes into account processes in three phases: the gas phase, inorganic salt-seeded aqueous phase, and dust phase, implying that uptake is treated as absorption (according to Henry's law). However, for many studies involving the uptake of traces gases on mineral dust, a Langmuir type behavior has been reported, showing an adsorption behavior that could typically contradict the assumption of absorption. Also, water and several gases have been showing to exhibit competitive adsorption properties, going against the absorption assumption. Could it not be that this two assumption would correspond to two completely different humidity regimes? Maybe the authors could comment on that, and strengthen their assumptions in the manuscript.

**Response:** The path for uptake of gaseous species onto dust particles changes depending on the environmental conditions. For the dry dust particles at very low humidity (less than 20% RH), the uptake of trace gases may follow the adsorption and desorption processes. Gustafsson et al. (2005) reported that ATD particles showed a considerably high affinity to water that the water content in ATD particles, which was measured by the thermogravimetric method, ranged 2-4 monolayers based on the BET surface area under the ambient humidity (20%-80%). Therefore, our model

approach begins with the absorption mode. As dust particles ages by the reaction of dust components (e.g., $CaCO_3$ and $MgCO_3$) with nitric acid, dust particles become even more hygroscopic (2 times higher than fresh dust).

In order to respond to the reviewer, we add the explanation of our assumption with absorption mode in the 1st paragraph of Sect. 3.1 and reads now, "Under ambient conditions (RH higher than 20%), studies showed that the water content in dust particles ranged 2-4 monolayers based on the BET surface area (Gustafsson et al., 2005; Yu et al., 2017). Therefore, we assume that the gas–dust partitioning of trace gases is governed by the absorption process."

**Comment 2:** Concerning the determination of the photoactivation parameters, have you checked how much is simply due to bleaching of the dye? That is a commonly reported issue in photocatalytic degradation of dyes on $TiO_2$.

**Response:** Figure S3 showed the degradation of the dye with and without dust. In the absence of dust particles, the decay of the dye was negligible (Fig. S3(c)). The sentence is added to the Sect 2.3 and reads, "The degradation of dye was significant only in the presence of dust particles".

**Comment 3:** In addition, the organic compound may also directly react with the electron-hole pair changing the reaction mechanism given in (R2) to (R5). Do you have any indication that this is occurring with the selected dye?

**Response:** We agree with the reviewer that the dye compound on dust may react with electrons-hole pairs. However, the amount of dye that is coated on dust particles (<1 µg per 200 µg of dust) is much smaller than water content (~ 50% of dry dust mass at 50% RH) on dust particles. Additionally, dust contacts with abundant oxygen molecules at the interface between gas and dust surface. If the primary process of the degradation of the dye is the reaction with an electron-hole pair, the degradation of dye is independent of humidity. As shown in Fig. 3, the estimated photoactivation parameters of both GDD and ATD particles increase with increasing humidity suggesting the importance of the role of water molecules to oxidize dye molecules. This explanation was also added to the end of Sect. 3.3 and reads now, "Additionally, the estimated

photoactivation parameters of both GDD and ATD particles increase with increasing humidity suggesting the importance of the role of water molecules to heterogeneous oxidation reactions."

**Comment 4:** did you performed any elemental analysis of the two samples, or just for the GDD as ATD has a known composition? This should then provide a basis for explaining the difference in the photoactivation parameters.

**Response:** Please also find the response to the comment 1 from referee #1. The fraction of elements in the GDD and ATD samples were previously analyzed by Park et al. (2017) using energy dispersive spectroscopy (EDS). The measured fractions of Fe and Ti in GDD is noticeably higher than that in ATD, which may explain GDD's higher photoactivation ability. The correlation of photoactivation ability and the dust metal composition needs to be parameterized in future.

**Comment 5:** Sulfate is very often considered as a poison for surfaces, as it passivates very rapidly reactive surfaces. However, the outcome of the AMAR does show that (probably due to the absorption assumption discussed above). Have you built in some capacity to have surface saturation or not at all?

**Response:** We have thought about this issue. We also think that sulfuric acid coating can damage photocatalytic ability of dust due to the reaction with conductive metal oxides (e.g., iron oxide and tritium oxide). However, the modification of photocatalytic ability of dust may needs great amounts of sulfuric acid and reaction time. In general, chamber experiments are conducted in high concentrations (~100 ppb $SO_2$, ~40 ppb $NO_x$ and ~400 µg cm$^{-3}$ GDD for 10 consecutive hours) compared to the typical ambient condition (20 ppb $SO_2$ even for highly polluted urban areas). We did not observe the reduction in photoactivation ability by coating dust with sulfuric acid during our chamber experiments. In order to estimate the length of time to yield the similar amount of sulfate with chamber-generated sulfate, we simulate sulfate formation using AMAR model under the polluted ambient condition (20 ppb of initial $SO_2$, 40 ppb $NO_x$ and 400 µg cm$^{-3}$ of GDD under environmental conditions at January 13, 2016). Our calculation shows that it takes 5-6 days. The reported average lifetime of airborne dust particles is ~4.3 days (S. et al., 2004;Scheuvens and Kandler, 2014), though their lifetime varies with particle size. Hence, the most dust particles

possibly settle down before they are significantly damaged by sulfuric acid coating. We conclude that dust's photocatalytic ability may not significantly changed during atmospheric aging. This explanation is added to the end of the 1$^{st}$ paragraph of Sect.5 Atmospheric implications and reads now.

"It is known that Inorganic acids can corrode metal oxides, but they first react with alkaline carbonates on dust. Additionally, the excess amount of sulfuric acid beyond the dust buffering capacity can be titrated by ammonia, which is ubiquitous in ambient air. Thus, the acidity of dust particles may not be high enough to damage the photocatalytic ability of mineral dust particles under ambient conditions. Based on our simulation (Fig. S9), it takes 5-6 days under the ambient conditions to produce the similar amount of sulfate observed in chamber studies (Fig. 4) and it is even longer than the reported average lifetime of airborne dust particles (~4.3 days) (S. et al., 2004;Scheuvens and Kandler, 2014). Therefore, most dust particles possibly would settle down before they are significantly corroded by sulfuric acid coating."

**Comment 6:** Figure 4 seems to show that the model does capture the nitrate formation at longer times. How do you explain this? Is there any renoxification process taking place in this system?

**Response:** In the AMAR model, the uptake of $HNO_3$ on dust is controlled by both gas-dust partitioning and heterogeneous reactions. In general, $HNO_3$ is abundant in urban areas due to high concentration of $NO_x$. The gaseous concentration of volatile $HNO_3$ (63.1 mmHg at 25 C°) is much higher than that needed for buffering dust. By this credential, nitrate salt is quickly regenerated even with the condition that nitrate is decomposed by renoxification. Thus, nitrate on the dust phase will be depleted only when alkaline cations react with other acids, which have the lower volatility than nitric acid.

**Reference:**

Gustafsson, R. J., Orlov, A., Badger, C. L., Griffiths, P. T., Cox, R. A., and Lambert, R. M.: A comprehensive evaluation of water uptake on atmospherically relevant mineral surfaces: DRIFT spectroscopy, thermogravimetric analysis and aerosol growth measurements, Atmos Chem Phys, 5, 3415-3421, DOI 10.5194/acp-5-3415-2005, 2005.

Park, J., Jang, M., and Yu, Z.: Heterogeneous Photo-oxidation of $SO_2$ in the Presence of Two Different Mineral Dust Particles: Gobi and Arizona Dust, Environ Sci Technol, 51, 9605-9613, 10.1021/acs.est.7b00588, 2017.

S., Z. C., L., M. R. L. R., and I., T.: Quantifying mineral dust mass budgets:Terminology, constraints, and current estimates, Eos, Transactions American Geophysical Union, 85, 509-512, doi:10.1029/2004EO480002, 2004.

Scheuvens, D., and Kandler, K.: On Composition, Morphology, and Size Distribution of Airborne Mineral Dust, in: Mineral Dust: A Key Player in the Earth System, edited by: Knippertz, P., and Stuut, J.-B. W., Springer Netherlands, Dordrecht, 15-49, 2014.

Yu, Z., Jang, M., and Park, J.: Modeling atmospheric mineral aerosol chemistry to predict heterogeneous photooxidation of $SO_2$, Atmos Chem Phys, 17, 10001-10017, 2017.

---

## Author Response (AR2)

**(Manuscript Ref. NO.: acp-2018-68)**

**Response to the comments from the Editor and the reviewers**

We appreciate the reviewers for their constructive comments on this manuscript. The detailed response to the comments has been listed below.

**Overall comments from the Editor:**

Thank you for your revision. The reviewers feel that their comments were largely addressed. One however still remains critical about one of the basic assumptions in the manuscript (i.e., absorption is taking place at the dust surface following Henry's law). I agree with this concern and also feel that the answer provided is weak in this regard. I strongly suggest that you attempt to more deeply address this issue in a revision.

**Comment from Referee #2:**

The authors have thoroughly revised their manuscript taking into account all comments raised by the reviewers. Nevertheless I am still puzzled by the assumption that the gas-dust particles interactions can be description by an absorption processes governed by Henry's law. This would mean that the surface concentration increases linearly with increasing gas phase concentration, which would not fit quite a few observations where a Langmuir-Hinshelwood type behavior has been observed. I acknowledge that the response of the authors that water may form multilayers on top of dust, but this may not confer to it the feature of bulk water. Can you with your assumptions reproduce surface processes as described by a Langmuir-Hinshelwood type equation? If yes, I would then be happy to recommend the publication of your paper.

**Response to reviewers:**

Our study mainly focused on authentic mineral dust particles (Gobi Desert Dust and Arizona Test dust) under the typical ambient relative humidity (RH) ranging from 20% to 80%. Under the very dry condition, we agree to the reviewers in that the sorption of trace gases and water molecules on mineral dust particles may be governed by a Langmuir-Hinshelwood type adsorption process. However, the observed hygroscopicity suggested that dust particles were coated with the multilayer of water under the RH higher than 20%. Thus, the sorption of trace gases was

approached by the absorption-base model frame (Fig. 2). Several studies also support our assumption showing that the uptake coefficients of trace gases (i.e., $SO_2$, ozone, and $H_2O_2$) on dust particles were positively sensitive to RH (0%-90%) (Vlasenko et al., 2006;Yang et al., 2017;Pradhan et al., 2010). Besides, the modification of dust surfaces by forming alkali nitrates can greatly increase hygroscopicity of dust as well as heterogeneous reactions of trace gases.

In order to response to the reviewer, the following sentences have been added to the beginning of Section 3.2 and reads now, "
[revised manuscript text omitted]